

# Mechanisms behind bottom-up effects: eutrophication increases fecundity by shortening the interspawning interval in stickleback

Anne Saarinen and Ulrika Candolin

Organismal and Evolutionary Biology, University of Helsinki, Helsinki, Finland

## ABSTRACT

Anthropogenic eutrophication is altering aquatic environments by promoting primary production. This influences the population dynamics of consumers through bottom-up effects, but the underlying mechanisms and pathways are not always clear. To evaluate and mitigate effects of eutrophication on ecological communities, more research is needed on the underlying factors. Here we show that anthropogenic eutrophication increases population fecundity in the threespine stickleback (*Gasterosteus aculeatus*) by increasing the number of times females reproduce—lifetime fecundity—rather than instantaneous fecundity. When we exposed females to nutrient-enriched waters with enhanced algal growth, their interspawning interval shortened but the size of their egg clutches, or the size of their eggs, did not change. The shortening of the interspawning interval was probably caused by higher food intake, as algae growth promotes the growth of preferred prey populations. Enhanced female lifetime fecundity could increase offspring production and, hence, influence population dynamics. In support of this, earlier studies show that more offspring are emerging in habitats with denser algae growth. Thus, our results stress the importance of considering lifetime fecundity, in addition to instantaneous fecundity, when investigating the impact of human-induced eutrophication on population processes. At a broader level, our results highlight the importance of following individuals over longer time spans when evaluating the pathways and processes through which environmental changes influence individual fitness and population processes.

# INTRODUCTION

Human activities are altering habitats around the world at an unprecedented rate and scale. A growing human-induced problem in aquatic ecosystems is eutrophication, the enrichment of ecosystems with nutrients (*Smith, 2003*). It promotes primary production, which in turn has a range of secondary effects on ecosystems, such as the accumulation of decomposing organic matter that uses up oxygen, the promotion of toxic algal blooms, and the proliferation of ephemeral algae in coastal systems (*Le Moal et al., 2019*; *Paerl et al., 2014*). These changes cause the decline of some species, while others profit and increase in

Corresponding author
Ulrika Candolin,
ulrika.candolin@helsinki.fi

abundance (*Levin et al., 2009*; *O'Neil et al., 2012*). Yet, the factors that determine whether a species profits or suffers from eutrophication are poorly known (*Di Carvalho & Wickham, 2019*; *Rigosi et al., 2014*)). This gap in our knowledge is due to the intricate effects of eutrophication on species—it influences species not only directly but also indirectly through ecological interactions and feedbacks among species (*Candolin, Bertell & Kallio, 2018*; *Hoover & Tylianakis, 2012*; *Wootton, 1994*). In addition, other human-induced disturbances can modify the effects of eutrophication, such as climate change (*Buma, 2015*; *Jackson et al., 2016*; *Russell et al., 2009*), coastal construction (*Kraufvelin et al., 2010*) and mesopredator release (*Kraufvelin, Christie & Gitmark, 2020*; *Ö et al., 2016*). Thus, to unravel the effects of eutrophication on species, the underlying mechanisms and pathways need to be identified and delineated.

In the Baltic Sea, nutrient concentrations started to increase in the 1950s because of human activities, reached a top in the 1980s and 1990s, after which the concentrations stabilised and even declined in many parts of the sea (*Andersen et al., 2017*; *Gustafsson et al., 2012*). However, in the Gulf of Finland, nutrient concentrations have continued to be high (*Andersen et al., 2017*). In addition, a range of other perturbations have occurred in the area that could interact with eutrophication, such as climate change (particularly warming), the overfishing of top predators, and the invasion of alien species (*Casini et al., 2008*; *Meier et al., 2019*; *Ojaveer et al., 2010*; *Rutgersson et al., 2014*). Yet, how these different disturbances influence biodiversity, directly and indirectly, and their individual and combined effects, are poorly known (*Meier et al., 2019*; *Vehmaa, Katajisto & Candolin, 2018*). This is hampering our ability to evaluate and mitigate the effects of eutrophication on the ecosystem, calling for more research into the topic.

The abundance of the threespine stickleback, *Gasterosteus aculeatus*, has increased in the Gulf of Finland during the last decades, as well as in other parts of the Baltic Sea (*Bergström et al., 2015*; *Candolin, Tukiainen & Bertell, 2016b*; *Candolin & Voigt, 2020*; *Olsson et al., 2019*). The increase has been attributed to a top-down effect from the decline of top-predators (*Eriksson et al., 2011*; *Ljunggren et al., 2010*). However, recent research indicates that a top-down effect alone cannot explain the increase, but that other factors must have contributed (*Bergström et al., 2015*). One such factor could be eutrophication and a bottom-up effect through increased algal growth and prey abundance. This is plausible as the threespine stickleback spends the summer in shallow coastal waters where eutrophication has enhanced the growth of filamentous algae (*Bäck, Lehvo & Blomster, 2000*; *Candolin, 2004*; *Gubelit & Kovalchuk, 2010*; *Kraufvelin et al., 2006*; *Rinne et al., 2018*; *Rinne, Salovius-Lauren & Mattila, 2011*) and thereby the abundance of prey, such as gammarids and other grazers (*Candolin, Johanson & Budria, 2016a*; *Olafsson et al., 2013*; *Salovius & Kraufvelin, 2004*), the preferred prey of the threespine stickleback in coastal waters (*Candolin, Johanson & Budria, 2016a*; *Jakubaviciute et al., 2017*). In support of a bottom-up effect, the population growth of the stickleback in the Gulf of Finland correlates with increased population fecundity in terms of the proportion of gravid females in the population (*Candolin & Voigt, 2020*). Females develop several sequential clutches of eggs during the breeding season and could profit from eutrophication in terms of increased food intake and, thus, fecundity. Males establish territories and build nests to which they

attract several females to spawn, while females spend most of their time foraging (*Wootton, 1984*). After spawning, the female leaves the nest and the male alone cares for the eggs until hatching (*Wootton, 1984*). Males do not benefit from increased food abundance in terms of improved body condition (*Candolin & Voigt, 2020*), as they reduce feeding while engaged in nesting activities (*Wootton, 1984*). Instead, males benefit from denser algal growth in term of shelter from predators and less interference from nest destroying and eggs stealing conspecifics (*Candolin, 2004*; *Candolin, Nieminen & Nyman, 2014*; *Candolin & Voigt, 1998*). Thus, improved male parenting ability in eutrophied habitats could ensure that the increased female fecundity results in increased offspring production.

The possibility that increased female fecundity has promoted offspring production in eutrophied habitats is supported by an increased number of fry emerging in spawning areas with a denser growth of algae (*Candolin, Nieminen & Nyman, 2014*). The increased offspring production could have contributed to the growth of the population, depending on the survival of the fry to the adult stage. However, whether such a bottom-up effect occurs—through increased female fecundity—is still an open question. Yet, determining the causes of the population growth is of paramount importance considering the key role that this mesopredator plays in the ecosystem. It influences the abundance of other species, both directly and indirectly, and affects ecological process such as energy flow and nutrient cycling (*Candolin, 2019*; *Des Roches et al., 2013*; *Jakobsen et al., 2003*; *Limberger et al., 2019*). Thus, more information is needed on the factors and processes that regulate the abundance of stickleback to improve our understanding of the causes and consequence of the changes occurring in the ecosystem, as well as to develop mitigation and adaptation strategies to minimize risks of large-scale changes.

We investigated if eutrophication increases female fecundity during the spawning season and, thus, if eutrophication could increase offspring production. To determine the effect of eutrophication on fecundity, we exposed females to high and low nutrient levels during the breeding season—which increases algae growth and prey availability—and recorded effects on the number and size of egg clutches produced.

## MATERIAL AND METHODS

We caught threespine stickleback before the breeding season in early May from a bay in the outer archipelago of the Northern Baltic Proper (59°50 ′N, 23°15′E) using minnow traps. We housed the fish in large holding tanks at a photoperiod of 18L:6D, a temperature of 18 °C, and a salinity of 0.6% to reflect natural conditions in the field. We moved females showing signs of sexual maturation (swollen abdomens and darker stripes on their lateral sides) to experimental tanks (28 l), one female to each tank, to follow their development of eggs. Before transfer, we measured their standard body length (± one mm), and after the first spawning event, their body weight (± 0.1 g). Body weight was not recorded before spawning as handling can cause females to release their eggs. The tanks were kept under natural light and temperature conditions, in several parallel lines on an outdoor platform. To ensure a healthy environment, sea water was slowly flowing through the tanks (about 100 ml/min) between 2 and 4 pm each day. All tanks received fresh sea water that had not

been through other experimental tanks. A net at the outflow (mesh size 0.2 mm) prevented organisms from escaping the tanks. The net was cleaned from algae every morning and evening.

To simulate habitats differing in eutrophication state and, thus, in the density of filamentous algae and associated prey, we exposed the females to one of two treatments: low or high algae density, with 50 g or 150 g respectively of the filamentous alga *Cladophora glomerata* evenly distributed over the bottom of the tanks, together with associated natural invertebrate fauna. The densities correspond to natural low and high algae densities in the field (*Candolin, Johanson & Budria, 2016a*). We manipulated algae and prey densities directly, rather than allowing algae and prey populations to gradually grow under two different nutrient levels, because of time constraints. The algae, with invertebrates, had been collected from the same area as the stickleback, and the algal filaments attached to plastic nets (35× 30 cm, mesh size one cm) at the bottom of the tanks through sewing. Before being placed in the tanks, the nets with algae had been held in a large common pool for one day to homogenize the distribution of invertebrates among the nets. Some grazers could have escaped during collection and transfer, and we consequently added five *Gammarus* sp. (about 10 mm) to the low-algae density treatment, and 15 *Gammarus* sp. to the high-algae density treatment, i.e., three times more prey to the tank with three times more algae. The additions were based on prior investigations of the amount of prey in the algae and the number lost when handling the algae, with the combined abundance of algae and prey reflecting conditions in high and low eutrophied habitats (*Candolin, Johanson & Budria, 2016a*). To ensure that the difference in algae density was maintained throughout the experiment, we added nutrients to the high-algae density tanks, by adding nutrients to the water flowing through the tanks during the 2 h of flow through (a nutrient mixture containing 0.046 g of nitrogen/l and 0.036 g of phosphorous/l), while no nutrients were added to the low-algae density tanks. The nutrient addition should not influence the stickleback directly, only indirectly through increased algae and prey growth, as stickleback do not feed on nutrients. The used nutrient concentrations have been found to promote algal growth in prior research (*Heuschele et al., 2009*; *Järvenpää & Lindström, 2004*). Sea water naturally contains nutrients, which maintained the algal growth at natural low levels in the low-algae tanks.

We observed the females daily to record their fecundity state. As soon as a female was ready to spawn, based on a distended belly and a dilated urogenital papilla, we allowed her to spawn her eggs in the nest of a male. Males had been collected from the same site as the females, and allowed to build nests in individual tanks (10 l), following methods in *Candolin (1997)*. When the male had fertilised the eggs, we removed both the male and the female and returned the female to her tank to continue to develop eggs. We collected the eggs from the nest 2 h later when the eggs had hardened, and counted the number in the clutch. To measure egg size, we measured the diameter of 10 eggs using a microscope with a micrometre eyepiece. Females release all ovulated eggs during a spawning (*Wootton, 1984*). We recorded female body weight after each spawning.

We allowed the females to produce as many egg clutches as they could until the end of July when the breeding season ended. All but five females died towards the end of

**Table 1** **The impact of eutrophication and the density of filamentous algae on the production of eggs, loss of body weight, and time in the experiment, of threespine stickleback females.**

| | Low algae Mean ± SE | High algae Mean ± SE | $F_{1,38}$ | P |
|---|---|---|---|---|
| Number of breeding cycles | 3.6 ± 0.2 | 5.0 ± 0.3 | 15.48 | <0.001 |
| Interbreeding interval, days | 19.1 ± 1.0 | 15.5 ± 0.8 | 7.60 | 0.009 |
| Clutch size, egg number | 153.6 ± 6.0 | 149.2 ± 6.0 | 0.27 | 0.60 |
| Egg diameter, mm | 1.77 ± 0.02 | 1.76 ± 0.02 | 0.11 | 0.74 |
| Loss of weight after spawning, % | 9.45 ± 1.49 | 7.85 ± 1.00 | 0.80 | 0.38 |
| Total loss of weight, % | 9.10 ± 3.42 | 7.70 ± 0.93 | 0.16 | 0.70 |
| Experimental days | 61.6 ± 2.69 | 66.25 ± 2.38 | 1.64 | 0.21 |

**Notes.**
ANOVA was used to analyse the data.

the experiment (three survived in the low-algae treatment, and two in the high-algae treatment), as stickleback in the present population breed during one breeding season after which they die (*Candolin, 2000*). We recorded mean interbreeding interval (days), number of egg clutches produced, mean number of eggs in each clutch, and mean diameter of eggs in each clutch. To measure the amount of zooplankton left in the tank at the end of the experiment, we took a 1 l water sample, filtered it through a 200 μm mesh net, and counted the number of zooplankton under a stereo microscope. To measure the amount of filamentous algae left in the tank, we detached the algae from the net and measured the wet weight. To measure the amount of benthos left in the algae, we collected all invertebrates visible to the eye (mainly amphipods, chironomids, gastropods, isopods, and caddisfly larvae), dried them on blotting paper, and measured their total weight.

We used 20 females per each treatment. We used ANOVA to analyse the data and checked the assumptions of the test before analyses. Females in the two treatments did not differ in body length at the start of the experiment ($F_{1,38} = 0.04$, $P = 0.85$) or in body weight after the first spawning ($F_{1,38} = 0.04$, $P = 0.84$).

The study was conducted according to national guidelines and meets the terms of the regional ethics committee, the National Animal Experimental Board in Finland.

## RESULTS

Females in the high-algae treatment completed more breeding cycles and had shorter interbreeding intervals than females in the low-algae treatment (Table 1). The two groups did not differ in the number of eggs in the clutches or in the size of the eggs (Table 1).

Females in the two algae treatments did not differ in the loss of body weight after each breeding cycle, or over the experimental days (Table 1), or in the number of days in the experiment (Table 1).

The amount of filamentous algae in the tanks at the end of the experiment was higher in the high-algae treatment (Table 2). Similarly, the amount of prey organisms was higher in the high-algae treatment at the end of the experiment, for both zooplankton and benthos (Table 2).

**Table 2** Conditions in the experimental tanks at the end of the experiment.

| | Low density Mean ± SE | High density Mean ± SE | $F_{1,38}$ | P |
|---|---|---|---|---|
| Filamentous algae, g | 69.50 ± 3.02 | 175.96 ± 4.98 | 333.65 | <0.001 |
| Number of zooplankton $l^{-1}$ | 24.15 ± 2.82 | 79.00 ± 8.94 | 34.22 | <0.001 |
| Benthos, g | 0.80 ± 0.28 | 2.91 ± 0.49 | 14.21 | 0.001 |

**Notes.**
ANOVA was used to analyse the data.

## DISCUSSION

Our high-algae density treatment increased the lifetime fecundity of threespine stickleback females, but did not change their instantaneous fecundity. Females produced more egg clutches at higher algae density by shortening their interspawning interval, but they did not alter the size of their egg clutches or the size of their eggs. Because females in the present population complete only one breeding season after which most of them die (*Candolin, 2000*), as was also found in this experiment, the produced egg clutches represent the lifetime egg production of the females.

The increase in egg production rate was most likely caused by the increased prey availability and, thus, higher food intake, as the abundance of prey was higher in denser algae growth. High density of algae promotes the population growth of many benthic organisms, such as gammarids (*Kraufvelin, 2007*; *Kraufvelin & Salovius, 2004*; *Kraufvelin et al., 2006*; *Olafsson et al., 2013*; *Salovius & Kraufvelin, 2004*), which are the dominant prey species of the stickleback in coastal waters (*Candolin, Johanson & Budria, 2016a*; *Jakubaviciute et al., 2017*). In support of an impact of increased prey availability on lifetime fecundity, earlier laboratory studies have found increased food ration to shorten the interspawning interval of females and increase the number of egg clutches they produce (*Ali & Wootton, 1999*; *Fletcher & Wootton, 1995*; *Wootton & Fletcher, 2009*). However, the interspawning intervals in the earlier laboratory studies—which used enchytraeid worms as food—were much shorter (about 4 days at high food ration, fig 2 in *Ali & Wootton (1999)* than in our mesocosm experiment with natural prey and algae densities (15.5 days on average). This indicates that females in our experiment were not producing eggs at their maximum capacity. Limited prey availability and energy spent on searching for, catching, handling, and digesting the natural prey probably limited egg production rate.

The cause of the lack of an effect of the high-algae treatment on clutch size could be physiological and morphological restrictions of females. Moreover, evolutionary processes could have favoured the production of several smaller clutches rather than a few large ones in order to spread out the risk of failed reproduction. Males may cannibalise eggs laid by females (*Candolin & Vlieger, 2013*), and producing many sequential egg clutches—laid at different times into different nests—may spread out the risk of egg cannibalism, and, hence, be a more successful strategy than spawning many eggs into a few nests.

An alternative, not mutually exclusive, explanation for the increased female lifetime fecundity compared to the increase prey abundance is that females felt safer in denser algal growth and, hence, experienced lower stress levels. This could have reduced energy

use and allowed more energy to be allocated to egg production. In support of this, dense algal growth decreases aggressive interactions among individuals (*Candolin, Nieminen & Nyman, 2014*) and reduces perceived predation risk from piscivorous fishes and birds (*Ajemian, Sohel & Mattila, 2015*; *Candolin & Voigt, 2001*; *Sohel & Lindström, 2015*). Thus, both increased energy intake and reduced energy use could have contributed to the increased lifetime fecundity. However, lower stress levels are unlikely to be the main cause of the increased lifetime fecundity, as females in high and low algae density did not differ in body condition after each spawning, or at the end of the spawning season. High stress levels in vertebrates have a stronger negative effect on body condition than on fecundity (*Crespi et al., 2013*), and no such effect was evident in our experiment. Moreover, both algae treatments reflected natural algae densities that the species has adapted to and, hence, should not constitute overly stressful conditions (*Candolin, 2004*). Yet, to decisively determine the relative contribution of increased energy intake and reduced energy use on lifetime fecundity, prey abundance and algae density need to be separately manipulated, which requires a much larger experiment than the present one.

The increased female lifetime fecundity could have further consequences for population dynamics by increasing offspring production. However, the effect on the population depends on the spawning success of the females and the survival of the offspring both pre- and posthatching. Earlier studies show that more nests are available and embryo survival is higher in denser algal growth (*Candolin, Nieminen & Nyman, 2014*), probably because of reduced aggressive interactions among parenting males and fewer intrusions by nest raiders and egg thieves (*Candolin & Vlieger, 2013*). Thus, more juveniles are emerging in densely vegetated habitats (*Candolin, Nieminen & Nyman, 2014*). However, whether the increased offspring production influences the growth of stickleback populations depends on density-dependent processes later in life, such as the risk of predation and the transmission of infections (*Murdoch, 1994*; *Rose et al., 2001*). More research is consequently needed before any form conclusions can be drawn about the impact of increased female fecundity on population growth.

The degree to which the documented effect of eutrophication on female lifetime fecundity reflects processes in other eutrophied habitats is unknown, as we are aware of no other longitudinal studies on lifetime fecundity of individual stickleback females. In populations where females breed during multiple years (*Wootton, 1984*), females could save resources for survival until the next breeding season rather than investing in multiple breeding cycles per season. Thus, an interesting question is whether females in population with predominantly one breeding season could alter their strategy towards investment into survival and multiple breeding seasons, through phenotypic plasticity or evolutionary changes. This could increase the size of the adult population—through extended longevity of females—and, hence, alter the dynamics of the population. Thus, different reproductive responses to eutrophication could, have different effects on population structure and dynamics.

The influence of eutrophication on populations varies among species, with some species benefitting while others are disfavoured, altering the species composition of communities (*Hossain et al., 2019*; *Jeppesen et al., 2005*; *Mehner et al., 2005*). The threespine

stickleback appears to be a winner in profiting from both increased offspring production (*Candolin, Nieminen & Nyman, 2014*) and reduced predation pressure (*Bergström et al., 2015*), possibly at the expense of other species (*Candolin, 2019*). For instance, the increased abundance of stickleback could reduce the abundance of piscivores, the predators of stickleback, as stickleback prey on the larvae of piscivores (*Byström et al., 2015*). This could further promote the decline of top predators (*Bergström et al., 2016*; *Casini et al., 2008*; *Ljunggren et al., 2010*) and result in a feedback loop where the decline of predators further promotes the growth of the stickleback population. Thus, changes in species abundances can have further consequences for ecological and evolutionary processes (*Candolin, 2019*; *Johannesson et al., 2011*; *Norkko et al., 2019*), which emphasises the importance of evaluating the effects of eutrophication on populations, including the mechanisms and pathways behind the effects.

## CONCLUSION

Our results show that eutrophication and increased algae growth enhances lifetime fecundity of threespine stickleback females, but not their instantaneous fecundity. The increased lifetime fecundity could have contributed to the increased offspring production in the investigated population, but whether this has contributed to the growth of the population deserves more investigations. More broadly, our results stress the importance of considering effects of eutrophication on lifetime fecundity—not only on instantaneous fecundity—when evaluating the impact of eutrophication on population dynamics and community composition.

## ACKNOWLEDGEMENTS

We thank Marita Kuusinen for assistance with the experiment.

### Funding
Financial support was provided by the Swedish Cultural Foundation (to Ulrika Candolin). The funders had no role in study design, data collection and analysis, decision to publish, or preparation of the manuscript.

### Grant Disclosures
The following grant information was disclosed by the authors:
Swedish Cultural Foundation.

### Competing Interests
The authors declare there are no competing interests.

### Author Contributions
- Anne Saarinen performed the experiments, analyzed the data, prepared figures and/or tables, authored or reviewed drafts of the paper, and approved the final draft.

- Ulrika Candolin conceived and designed the experiments, analyzed the data, prepared figures and/or tables, authored or reviewed drafts of the paper, and approved the final draft.

## Animal Ethics

The following information was supplied relating to ethical approvals (i.e., approving body and any reference numbers):

The study was conducted according to national guidelines and meets the terms of the regional ethics committee, the National Animal Experimental Board in Finland. This study was judged to belong to class 0 by the National Animal Experimental Board in Finland.

## Data Availability

Raw data are available in the Supplementary File.

## Supplemental Information

Supplemental information for this article can be found online at http://dx.doi.org/10.7717/peerj.9521#supplemental-information.

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
