# Peer review of "Mechanisms behind bottom-up effects: eutrophication increases fecundity by shortening the interspawning interval in stickleback"

_PeerJ, doi:10.7717/peerj.9521_

## Round 0.1 · original submission · Major Revisions

Although your manuscript is clearly written and understandable some major changes are necessaries in order to improve some methodological aspects and validity of the findings. Please see the reviewers' comments for more specific actions on the paper.

Reviewer 1 ·

Basic reporting

“Mechanisms behind bottom-up effects: 1 Eutrophication increases fecundity by shortening the interspawning interval in sticklebacks” is a informative study about the potential mechanisms of bottom up effects of eutrophication on population dynamics in sticklebacks.
The article is well structured, while there is no figure in the publication the included tables are easy to read and informative enough.
The authors deliver a good introduction reporting the different ways eutrophication can interact with community composition and dynamics of the ecosystem.

Experimental design

The authors conducted a lab experiment where they find that eutrophied conditions in the field with high amount of algae and food can favour shorter inter clutch intervals and thus lead to a higher female lifetime fecundity.

- The study itself uses 2 treatments, high algae + more food, and low algae + little food. The authors discuss that the observed the effects on lifetime fecundity could be due to the “protection” by algae (l194ff) or from an increased food availability (l180ff). While the two chosen treatments reflect the conditions in the field, it would have been great to include the other two combinations as well, to be able to discern the relative contribution of both. I can see that the workload is already very high with just the two treatments, but should maybe be mentioned in the discussion as a “blue print” for further studies. Or it could be indirectly addressed by trying to compare the observed change in interspawing interval to changes observed from experiments with varying feeding levels. (see general questions below)

Validity of the findings

The findings are valid, the results are not overly interpreted and shortcomings due to the experimental design are addressed.

Additional comments

Here are some general comments, questions, and minor corrections which I think would improve the manuscript:

- How does the reduction of the inter-spawning interval in this study compare to previous studies with varying food availability? I think this information could be useful to discuss how much of the effect could be due to food compared to stress relief.

- l 223, maybe add “…and an increased number of nests in dense vegetation”? I guess that is more or equally important than hatching success, especially since Stickleback males do not take as many eggs as possible, but “close” their nest after a while, hence females would be limited by nests to place their clutches in this limitation could be removed in eutrophied areas.

- One topic i was missing is how does predation risk change in eutrophied areas? do females or offspring get eaten more or less in eutrophied areas? While trying to scout the literature I came across this paper : Gagnon et al 2017 (Shifts in coastal fish communities: Is eutrophication always beneficial for sticklebacks? ) where the authors claim that eutrophication/filamentous algae are actually negatively correlated with stickleback abundance in the field, partly because there is less cover against predation in these areas. While I think their stats are a bit weak, it would maybe be good to discuss your results in perspective with theirs.

l37: shouldn’t it be rather: “the enrichment of ecosystems with nutrients?”

l56: ..”such as climate change” is it through warming or the predicted precipitation/freshwater inflow. It would be nice to be more specific here.

l61. “…calling for more research into the topic.“

l96 maybe specify:” the larvae of piscivores” to be clear that they are not eating their own.

l110 the description of the location could be more precise, at the moment it leads me to somewhere in the baltic outside of Helsinki.

l128 “thank” should be “tank”

l131. added nutrients to one but not the other treatment… Could this also have influenced the results? I cannot see any obvious connection, given that sticklebacks obviously do not feed on nutrients, but I think it deserves a mention…

Reviewer 2 ·

Basic reporting

'no comment'

Experimental design

l.102-106 - The aim should be more clearly presented. Remove "and an increased density of filamentous algae and prey abundance" (l.102) and consider this version: "We investigated if eutrophication increases female fecundity during the spawning season." as you explain the details of your treatment in the next sentence.

l.110 - please provide more exact coordinates

l.114 - was Standard or Total body length measured?

l.132-133 - please add information on what you actually added (nitrates and phosphates?)

l.136 - this is the only spot, where the "low" treatment is described as a control. For the overall perspective, consider making it more obvious throughout the text.
l.138 - fishes do not have cloaca, use urogenital papilla instead
l.157 - The raw data seems consistent with little intra-individual variation, so ANOVA on averaged values seems appropriate. I just wonder why the Authors did not use mixed-effects models with female ID as random factor to gain statistical power in their analyses and avoid losing the information on among-clutch variation.

l.157 - it seems worth to be more exact on the sample size here. I found that 20 females per treatment were analysed only after checking the original data file. Consider writing something like: "We used 20 females per each treatment". Under replicates, one can imagine individual spawning events too, but your study was much more robust, so don't be afraid to write that down. Also because of this strange wording, I wonder whether there was more females used in the experiment that did not reproduce. If so, this should be explained in the text along with their respective number in each treatment.

Validity of the findings

l.174 - I suggest to change "Nutrient enrichment and a high density of filamentous algae" to "Our high algae density treatment" as the manipulation was complex and also included adding invertebrate prey.
l.180-193 - Is there any information on what is the number of clutches in the natural population?
l.182 - I would rephrase "The enrichment with nutrients maintained high algae growth conditions across the season, which is known to" with simply "High density of algae" as it seems that algae relative growth was lower in High algae treatment compared to the low (Table 2 - (69.50-50)/50 vs. (175.96-150)/150).

Additional comments

The study by Anne Saarinen and Ulrika Candolin presents an interesting experimental approach to test the effect of eutrophication on female reproduction in three-spined stickleback. They manipulate amount of filamentous algae, number of invertebrates in the tank, and concentration of nutrients on tank inflow, to create high and low treatment. They find that despite the clutch and egg size remain similar between the two treatments, the high algae sticklebacks are able to spawn more frequently. The Authors then put their findings into the context of population dynamics under anthropogenic influence.
The manuscript reads well overall and is clearly structured. The aims, methods and results are concise and sufficiently detailed. Although the experimental design consists of a multivariate manipulation (higher amount of algae, adding Gammarus sp., and nutrients), the use of "High algae density" is a reasonable umbrella label and seems carefully handled and described. Data analysis was performed appropriately and the main outcomes are sound. Discussion provides reader with enough context of the current research on eutrophication effects on stickleback populations, and also offers potential explanations for the observed pattern. The study therefore suits publication in PeerJ and I suggest minor changes in addition to the comments provided in the Review sections above (1.-3.):

l.37 - maybe change to "with" in "of nutrients"
l.80 - remove "to" from "do not to"
l.210 - reword to "Whether this contributes to the growth of stickleback populations..."
l.222 - "hatching success of males" reads inappropriately and I would talk about increased recruitment or even nesting success.
Tables 1 & 2 - Label treatments as "High algae" and "Low algae". "High density" confuses readers.

Reviewer 3 ·

Basic reporting

The manuscript by Saarinen and Candolin aims to test the effects of the widespread phenomenon of eutrophication (as the anthropogenic increase of nutrients) on the life-cycle of three-spined sticklebacks, particularly focusing on its effects on female reproductive traits. Using a laboratory aquarium experiment, the results showed that eutrophication leads to higher lifetime fecundity of female stickleback either by altering the habitat structure (providing more shelter) and/or increasing the productivity of the system (having higher prey availability). The mechanism behind higher lifetime fecundity was due to an increase in the number of times the females spawned (via reducing the time between spawning events), rather than changes in clutch size, egg size or the invested energy to reproduction (based on changes in body weight).
Overall, the manuscript is clearly written and understandable, it provides context and niche to the study citing relevant works, presenting relevant tables and supplying raw data that is understandable.

Experimental design

In order to test the working hypothesis (whether eutrophication increase fecundity in female sticklebacks) the authors use an experimental design in which the authors compare control tanks with a tanks subjected to a eutrophication scenario by: (1) adding more algae, (2) adding more prey, (3) adding more nutrients. Despite all three variables are likely to come together during eutrophication, this multiple simultaneous manipulation makes a bit difficult to decipher the mechanism that affects fecundity in fish. A longer term incubation before the fish addition just with nutrient additions and equal initial conditions of algae and prey would have provided a better evidence of the collinearity of these variables. Please justify the simultaneous manipulation of so many variables between control and treatment aquaria.
In addition, as the authors state in the discussion (Lines 194-219), it is difficult to separate a shelter explanation from the increasing productivity explanation of the increasing fecundity. In future studies this can be solved in by manipulating the amount of shelter with artificial substrate (e.g. plastic brush piles) independently from nutrients or food. Since the mechanisms of the effects of eutrophication are difficult to disentangle, I would tone down all the statements of highlighting the impact of this study in understanding the mechanisms by which eutrophication affect fish populations (Lines 16-17, 49-51). Additionally, since it is not clear that productivity explained this pattern instead of shelter, I would advise not to use the bottom-up (productivity) effects but more eutrophication as a mediator of both productivity and habitat changes, so please reconsider to change the title of the article either adding habitat changes or removing bottom-up effects.

Validity of the findings

The implications of these results are important and relevant, but I would be cautious when upscaling these results to patterns in the whole population or even more to the community and ecosystem level. This study shows the effects of potential eutrophication only to one part of the population (females) and during a short (yet important) period of their life histories. How eutrophication affects the males, the actual mating success and the survival and growth of the sticklebacks during ontogeny is yet unclear based on this study and may affect the outcome of population dynamics. Yet, the mechanism here presented may explain (to some extent) the trends found in natural populations (Candolin and Voigt, 2020).

Additional comments

Specific comments
Abstract
Line 24-25: the results of more offspring emerging I habitats with denser algal growth are not presented in the study. Please make it clear that this knowledge comes from other studies or delete it.
Introduction
Lines 85-101: speculative sentences that would fit more the discussion than the introduction.
Materials and methods
Line 110-111: Motivate why using the temperature and salinity conditions (based on natural conditions?).
Line 113-114: Why weighting the females after the first spawning event and not before as well =?
Line 115-116: Could you explain how the tanks where distributed spatially. Did the same water flow through the treatment aquaria?
Line 117: I assume that the 0.2 mm net clogged with algal growth during the experiment, especially in the treatment enclosures. Did you do anything about it?

Line 118-121: Can you describe the spatial design within each enclosure?
Line 132-133: Which type of nitrogen and phosphorous were added? The addition of nitrogen and phosphorous are far away from natural proportions for optimal growth (Redfield ratio is 16/1 N/P molar, 7.4/1 by mass; 1.27 N/P by mass in this study ). Please specify why adding these N/P ratios and state which inorganic N and P was added (Chemical formula).
Line 161: please estate the number of the Ethical permit used.
Discussion conclusion
Line 178 Please provide reference to the statement that that population of sticklebacks dies after one breeding season. This raises the question, how representative is this strategy in sticklebacks? Is that widespread in other populations in the Baltic Sea or over the worls? Please try to add a few sentences comparing with other populations or fish species.
Line 191-193 please elaborate a bit more (add one sentence) on why is this advantageous.
References
Please be consistent when reporting references and pay special attention to the capitalization of words in titles. Only the first word of the title should be capitalized.

---

## Round 0.2 · accepted · Accept

Dear Authors,

I am pleased to confirm that your paper has been accepted for publication in PeerJ.

Thank you for submitting your work to this journal.

Reviewer 1 ·

Basic reporting

No change to my previous assessment. The study is clearly reported, the background is sufficient and gives a good overview over the topic. The raw data is available..

Experimental design

The authors conducted a lab experiment where they find that eutrophied conditions in the field with high amount of algae and food can favour shorter inter clutch intervals and thus lead to a higher female lifetime fecundity.

Validity of the findings

The findings are valid, the results are not overly interpreted and shortcomings due to the experimental design are addressed.

Additional comments

I am happy with the changes and the responses of the authors to my and also the comments of the other reviewers.

I only found two minor misspellings in the last version:
L 241 “…. to the increased prey ….” should be “…. to the increased prey ….” (?)
L 276 remove the second “,”

Reviewer 2 ·

Basic reporting

'no comment'

Experimental design

'no comment'

Validity of the findings

'no comment'

Additional comments

I think the Authors have put reasonable amount of effort to clarify their manuscript and I am happy with the changes made.

Reviewer 3 ·

Basic reporting

The authors have addressed all the comments raised succesfully. I therefore endorse the publication of the article.

Kind regards.

Experimental design

Experimental design is now clarified and supported in previous literature.

Validity of the findings

The findings are scientifically correct and implications are now toned down according to the level of evidence provided.